# The difference between cellulolytic 'culturomes' and microbiomes inhabiting two contrasting soil types

Elizaveta V. Evdokimova[1,2]☯*, Grigory V. Gladkov[1,2]☯, Natalya I. Kuzina[2], Ekaterina A. Ivanova[3], Anastasiia K. Kimeklis[1,2], Aleksei O. Zverev[1,2], Arina A. Kichko[1,2], Tatyana S. Aksenova[2], Alexander G. Pinaev[2], Evgeny E. Andronov[1,2,3]

**1** Saint-Petersburg State University, Saint-Petersburg, Russia, **2** All-Russia research institute for agricultural microbiology, Saint-Petersburg, Russia, **3** Dokuchaev Soil Institute, Moscow, Russia

☯ These authors contributed equally to this work.
* pershina.elizaveta@yandex.ru

## Abstract

High-throughput 16S rRNA sequencing was performed to compare the microbiomes inhabiting two contrasting soil types—sod-podzolic soil and chernozem—and the corresponding culturome communities of potentially cellulolytic bacteria cultured on standard Hutchinson media. For each soil type, soil-specific microorganisms have been identified: for sod-podzolic soil—*Acidothermus*, *Devosia*, *Phenylobacterium* and *Tumebacillus*, and for chernozem soil—*Sphingomonas*, *Bacillus* and *Blastococcus*. The dynamics of differences between soil types for bulk soil samples and culturomes varied depending on the taxonomic level of the corresponding phylotypes. At high taxonomic levels, the number of common taxa between soil types increased more slowly for bulk soil than for culturome. Differences between soil-specific phylotypes were detected in bulk soil at a low taxonomic level (genus, species). A total of 13 phylotypes were represented both in soil and in culturome. No relationship was shown between the abundance of these phylotypes in soil and culturome.

## Introduction

Cellulolytic microorganisms are one of the most popular subjects of scientific research for several reasons. On the one hand, they have an understandable pattern of nutritional requirements, which makes the selection of a nutritional medium more convenient. On the other hand, they play a crucial role in the process of soil formation and global carbon cycling. Modern metagenomic techniques provide new opportunities to investigate cellulolytic soil communities.

Five different North American forests were studied by Wilhelm with co-authors, who uncovered the biodiversity of lignocellulose-, hemicellulose- and cellulose-degrading bacteria and fungi in soil [1]. It was shown that bacteria from the Caulobacteraceae family were the most active decomposers and utilised all the proposed substrates. The cellulolytic component of the community was enriched with bacteria belonging to the Burkholderiaceae, Comamonadaceae and Oxalobacteraceae families. Cellulose and hemicellulose together were decomposed

**Data Availability Statement:** All illumina sequencing files are available from the Genbank database SRA (NCBI) within the dataset: SubmissionID: SUB5714186; BioProject ID: PRJNA549392

**Funding:** The work is supported by Russian Science Foundation (http://rscf.ru/en/) grant named RSF 18-16-00073. NO

**Competing interests:** The authors have declared that no competing interests exist.

mostly by bacteria belonging to the *Asticaccaulis*, *Cellvibrio*, *Janthinobacterium*, *Cytophaga* and *Salinibacterium* genera. In addition, many previously uncultivated bacteria from relatively new phyla were detected, particularly bacteria from the genera *Chtoniobacter*, *Opititus* (phylum Verrucomicrobia) and Candidatus *Saccharibacteria* (TM7) [1].

The characteristic genera of relatively cold temperate forests (e.g., pine forests and mountain pine forests) were also found by Štursová and co-authors. Among them were *Cytophaga*, *Pedobacter*, *Burkholderia*, Gp1 and Gp2 acidobacteria, *Asticaccaulis*, *Achromobacter*, *Mucilaginibacter*, *Herminiimonas*, *Collimonas* and others [2].

A study by Eichorst and Kuske [3] covered the biodiversity of soil bacteria in five different ecosystems within both temperate and subtropical climatic zones. In agreement with previous studies, a prominent role of the Caulobacteriaceae and Burkholderiaceae families in xylan and cellulose degradation was also discovered in this research. Additionally, the list of cellulolytic bacteria was completed with the families Rhizobiales, Sphingobacteriales, Xanthomonadales and *Myxococcales* as well as unidentified and still uncultured representatives of Acidobacteria group I [3].

Other aspects of metagenomic studies involve large-insert metagenomic library analysis [4], shotgun targeted metagenomic sequencing [5, 6], or proteomics [7] and aim to study not only the taxonomic structure of the cellulolytic community but also the biodiversity of genes involved in decomposition. However, it is not necessary to dwell on them here, because their goals are beyond the scope of the current study.

The only drawback of metagenomic analysis is its dissociation from conventional microbiological methods, which in turn can provide comprehensive information on the physiology of the mineralisation processes in microbial cells. Sharing of data obtained using these methods simultaneously will update the available information on already cultivated microorganisms and emphasise the promising species that should be cultivated for future research.

Currently, there are plenty of works aimed at the comparative analysis of metagenomic data and various culturing methods. However, soil microbiology is still lacking examples of this kind of study. The application of these methods leads to the revision of the composition of the cultivated part of the soil, or even the worldwide microbiome. For example, Shade with co-authors demonstrated that a nutritional medium usually captures only a tiny part of the rhizosphere microbial community, containing in turn the minor phylotypes from the corresponding metagenomic library. Moreover, some species that were detected on a medium did not have corresponding signatures in the metagenomic analysis [8]. This study was conducted by using a very rich medium (specifically the rhizosphere isolation medium, RIM, which contains glucose, amino-acid mixture and vitamins, [9]), so the presence of many spore-forming copiotrophs could introduce significant inaccuracy in the biodiversity data. In this work, we used a relatively poor medium, in which cellulose filter paper was the only source of available carbon, so this should produce more selective conditions for bacterial growth than those in the aforementioned study.

Despite the apparent simplicity of the culturing procedure for cellulolytic bacteria, there is still a lack of comprehensive studies devoted to the biodiversity of microbiomes inhabiting several climatic zones. Known studies with similar goals focused on microbiomes of the Brazilian mangroves [10], sugarcane plots in Mexico [11], or paddy fields in Hainan [12], which all are very specific biomes. Considering the wide fluctuations in the composition of the soil microbiome depending on climate changes [13], we still have much to explore in terms of the biodiversity of the microbial consortia inhabiting climatically unstable subtropical and temperate regions. Thus, the main goal for the current study is simultaneous analysis of cellulolytic culturomes and the corresponding metagenomes in two contrasting biomes—sod-podzolic and chernozem soils—from the temperate and subtropical grasslands of Russia.

## Materials and methods

### Soil sample collection

Soil samples of sod-podzolic (SP) and chernozem (CZ) soil were collected in summer 2017 during expeditions to the Pskov (Pskov Research Institute of Agriculture, 57˚50'44.2"N, 28˚12'03.7"E) and Voronezh (Kamennaya Steppe reserve, 51˚01'41.6"N, 40˚43 '39.3"E) regions, respectively (**S1 Fig**). The director of Federal State Budget Scientific Institution "Kamennaya Steppe Experimental Forest District" and the director of the Federal State Budget Scientific Institution Pskov Research Institute of Agriculture gave permission for the sample collection. Soil samples were taken from the territories of the formerly sown areas from 10 different equidistant points from the upper soil layer (approximately 10 cm from the top of the soil profile). Finally, the selected samples were mixed and transported for laboratory research. Six replicates for each type of soil were formed.

### Bacterial growth on nutritional medium

For cultivation, solid Hutchinson medium [14] with cellulose filters was used (grams/L: $NaNO_3$: 2.5, $FeCl_3$: 0.01, $K_2HPO_4$: 1.0, $MgSO_4 \cdot 7H_2O$: 0.3, NaCl: 0.1 and $CaCl_2$: 0.1; pH 7.2). The analysis was performed in six replicates for each type of soil. For both soils, the active growth of various types of bacteria was detected. After two weeks of cultivation, Petri dishes were washed out with sterile water, centrifuged, and subjected to DNA isolation. These samples were named gSP and gCZ accordingly.

### DNA extraction and sequencing

The DNA was extracted from 0.2 g of soil using the PowerSoil DNA Isolation Kit (Mobio Laboratories, Solana Beach, CA, USA), which included a bead-beating step, according to the manufacturer's specifications. Samples were homogenised with a Precellys 24 (Bertin Corp., USA) at 6.5 m/sec, twice for 30 s. The purity and quantity of DNA were tested by electrophoresis in 1% agarose in $0.5 \times$ TAE buffer. DNA concentrations were measured at 260 nm using a SPECTROStar Nano (BMG LABTECH, Ortenberg, Germany).

The same DNA extraction procedure was applied to the culture plates. Microbial colonies were removed and solubilised in the extraction buffer, and DNA was extracted according to the manufacturer's instructions. The average DNA yield was 2–5 µg DNA, with concentrations between 30 and 50 ng/µl. The purified DNA templates were amplified with the universal multiplex primers F515 5′–GTGCCAGCMGCCGCGGTAA–3′ and R806 5′–GGACTACVSGGGTATC TAAT–3′ [15] targeting the variable region V4 of bacterial and archaeal 16S rRNA genes, flanking an approximately 300-bp fragment of the gene, extended with service sequences containing linkers and barcodes according to Illumina technology. The PCR reactions were assembled in a 15-µl mix containing 1 U of Phusion Hot Start II High-Fidelity polymerase and 1X Phusion buffer (Thermo Fisher Scientific, USA), 5 pM of both primers, 10 ng of DNA, and 2 nM of each dNTP (Life Technologies, USA). The PCR thermal profile used was 94˚C for 30 s, 50˚C for 30 s, and 72˚C for 30 s for 29 cycles. A final extension was performed at 72˚C for 3 min. PCR products were purified and size selected with AM Pure XP (Beckman Coulter, USA). Further library preparation was done according to the manufacturer's protocol with the MiSeq Reagent Kit Preparation Guide (Illumina, USA). Libraries were sequenced on an Illumina Miseq with a MiSeq® Reagent Kit v3 (2x300b) sequencing kit.

### Data processing

Amplicon libraries of the 16S rRNA gene were processed using packages in R [16] and QIIME2 [17] software environments. RStudio [18] was used as the development environment

for R. Raw sequence reads were trimmed and grouped into amplicon sequence variants (phylotypes) by use of the 'dada2' package [16]. The RDP classifier [19] based on Silva 132 [20] was used to classify assign taxonomic ranks to the phylotypes. The phylogenetic tree was built in the QIIME2 software environment in the SEPP package [21]. Data were normalised by a rarefaction algorithm according to the sample with the smallest number of readings for alpha and beta-diversity analysis. For differential analysis of phylotypes and quantitative metrics, the normalisation was performed by a variance stabilisation algorithm through the 'DEseq2' package [22]. To estimate the significance of differences between phylotypes previously normalised data were processed using the Wald test, with Benjamin-Hochberg false discovery rate (FDR) correction in the 'DEseq2' package [23]. The UniFrac, unweighted UniFrac [24], Bray-Curtis and MPD [25] algorithms were used as metrics for beta diversity. Beta-diversity data was graphically reproduced using PCoA [26]. Statistical analysis of beta-diversity was done by PERMANOVA [27] in the form of the *adonis2* function ('vegan' package) [28]. The formula by Apostol and Mnatsakanian [29] in package 'usedist' [30] was used as an additional statistical approach to calculate the distance between the centres of mass (centroids) of the sample groups in the beta-diversity space. The function *cophenetic.phylo* from the 'ape' [31] package was used to agglomerate closely related taxa using single-linkage clustering. The reliability of the dependence of the representation of phylotypes in soil and culturomes was obtained through the Fisher test for the generalised linear model ('glm') [32]. The R packages 'phyloseq' [33], 'ggpubr' [34], 'picante' [35], 'ggforce' [36], 'tidyverse' [37], 'ggtree' [38], 'ampvis2' [39] and 'rnaturalearth' [40] were used for post-processing and visualisation of the obtained data.

## Data deposition

All sequences were deposited to the SRA (NCBI) within the dataset: Submission ID: SUB5714186 and BioProject ID: PRJNA549392.

## Results

### Alpha diversity of soil microbiomes and culturomes

An amplicon library was obtained for bulk soil samples (246,527 sequences), and culturomes (397,307 sequences). Phylotype richness was higher in bulk soils compared to culturomes in both sample sets; sod-podzolic (SP) soil was more diverse than chernozem (CZ) soil (bulk soils: SP—1505, CZ—1286 and culturomes: SP—274, CZ—239, **Fig 1A**). These tendencies can be clearly seen on the rarefaction curves, where culturome samples reached plateaus much earlier than bulk soil samples. The alpha diversity indices (evenness and richness) were similar for bulk soils as well as for culturomes (SP—424, CZ—358 and SP—65, CZ—54 correspondingly). The complete description of vertical soil structure as well as the agrochemical analysis can be seen in Table 1.

### Identification of the core and accessory components of soil microbiomes and culturomes

53 phylotypes for SP and 39 phylotypes for CZ were shared between bulk soil and culturome samples (**S1 Table**). This set of phylotypes was dominated by Gammaproteobacteria (*Massilia*, *Pseudoduganella*), Actinobacteria (*Streptomyces*, *Glycomyces*, *Pseudarthrobacter*), Alphaproteobacteria (*Bradyrhizobium*, *Devosia*, *Microvirga*), Bacteroidetes (*Niastella*, *Dyadobacter*, *Chitinophaga* (predominated in SP)), Firmicutes (*Bacillus*(predominated in SP), *Paenibacillus*).No relationship was found between the representation of the phylotype in the bulk soil and culturome (**Fig 1B**; P-value for general linear F-test for SP was 0.43, for CZ– 0.5).

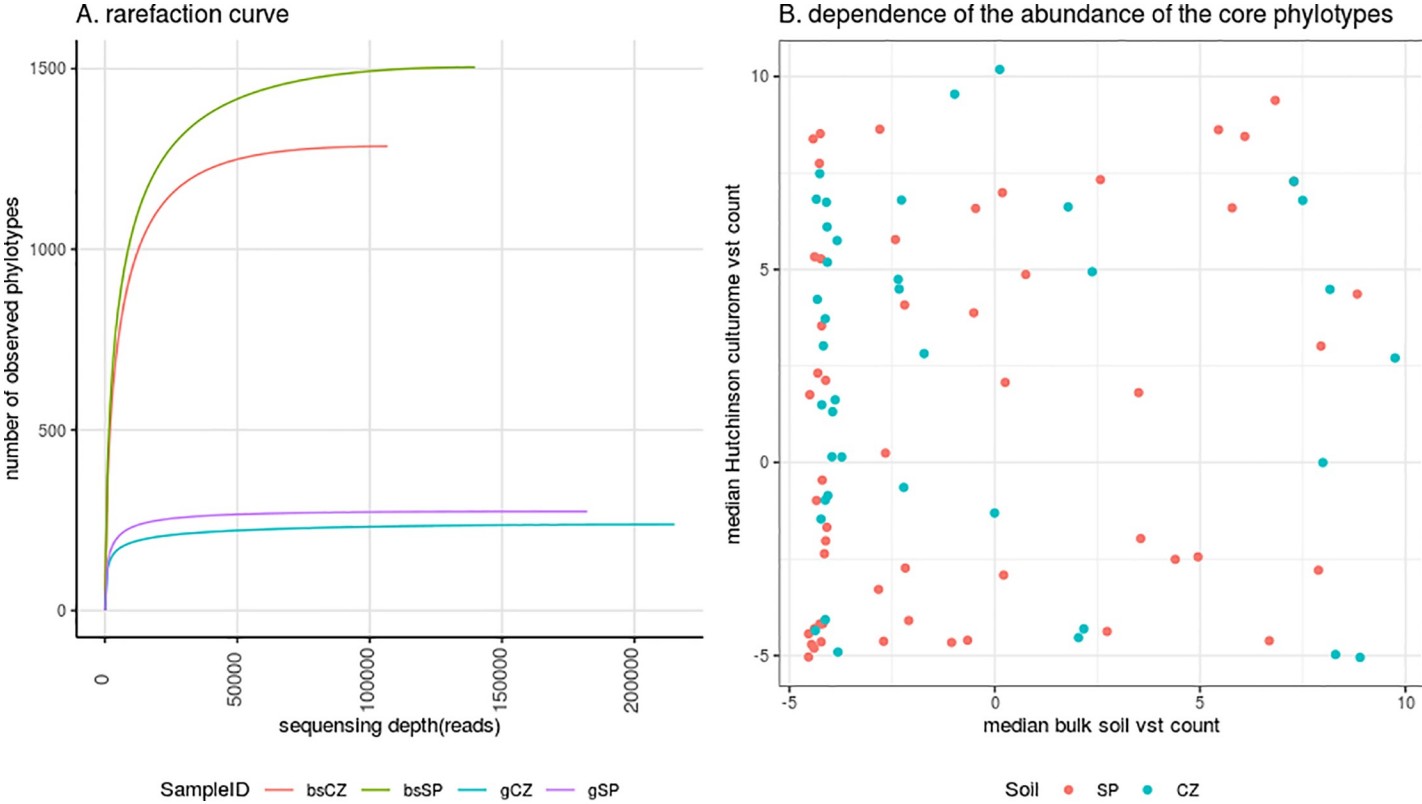

**Fig 1.** A. Rarefaction curves for the culturomes (gCZ—Hutchinson medium culturome from the chernozem soil, gSP—Hutchinson medium culturome from the sod-podzolic soil) and bulk soil samples (bsCZ—bulk chernozem soil, bsSP—bulk sod-podzolic soil) B. Dependence of phylotype abundance in the culturome (Y-axis) as compared to bulk soil (X-axis). Color indicates different soil types.

**Table 1. The main physical and agrochemical characteristics of the analyzed soil samples.**

| Parameter | Units | Soil type | |
|---|---|---|---|
| | | Sod-podzolic | Chernozem |
| Particle size (mm) distribution | | | |
| 1–0.25 | % | 15.1 | 1.8 |
| 0.25–0.10 | % | 16.4 | 1.1 |
| 0.10–0.05 | % | 23.26 | 10.86 |
| 0.05–0.01 | % | 28.64 | 33.36 |
| 0.01–0.005 | % | 1.8 | 7.68 |
| 0.005–0.001 | % | 3.2 | 11.08 |
| <0.001 | % | 11.6 | 34.12 |
| Agro-chemical analysis | | | |
| pH | pH units | 6.05 | 7.32 |
| N | % | 0.22 | 0.38 |
| P | mg/kg | 85 | 121 |
| K | mg/kg | 60 | 155 |
| C | % | 2.48 | 8.75 |
| Ca | mmol/100 g | 3.25 | 30.62 |
| Mg | mmol/100 g | 2.45 | 3.82 |

Soil specific taxa for bulk soil were phylotypes from the Verrucomicrobia (Candidatus *Udaeobacter* phylotypes), Actinobacteria (*Microlunatus* phylotypes, *Acidothermus* in SP, *Blastococcus* in CZ), Bacteroidetes (*Chitinophagaceae* phylotypes, *Phylobactius bibliophyllum* phylotypes) phylotypes in SP, *Bacillus* in CZ, Acidobacteria (RB41 phylotypes, *Bryobacter* phylotypes, *Candidatus_Solibacter* phylotypes in SP), Entotheonellaeota (specific for CZ) and Alphaproteobacteria (*Pseudolabrys* phylotypes, *Sphingobacteriales* phylotypes in CZ; **Fig 2B**). Significant increases of 295 phylotypes for SP and 213 for CZ were shown. Differences between soil specific phylotypes were manifested in bulk soil at a low taxonomic level (genus, species). Sod-podzolic soil as compared to the chernozem was characterised by the high values of the species abundance variance within a certain genus.

The number of the common phylotypes between the culturomes derived from two soil types was substantially lower compared to bulk soil samples. The culturomes were enriched with Proteobacteria and Actinobacteria and to a lesser extent with Firmicutes and Bacteroidetes. As opposed to bulk soil samples, in culturomes, the presence of the phylum Acidobacteria, Entotheonellaeota was not shown, and archaea were not represented. The representatives of the genus *Streptomyces* predominated among the detected actinobacteria. Verrucomicrobia was represented only by the Verrucomicrobiaceae family, while the representatives of Chthoniobacteraceae prevailed in the soil samples. The maximal relative abundances were detected for *Pseudoduganella*, *Pseudoxanthomonas*, *Massilia* (Gammaproteobacteria), *Streptomyces* and *Glycomyces* (Actinobacteria) in CZ and *Streptomyces* (Actinobacteria), *Chitinophaga* (Bacteroidetes), *Massilia aerilata*, *Variovorax paradoxus* and *Pseudomonas (*Gammaproteobacteria*)* in SP.

## Beta-diversity of soil microbiomes and culturomes

In both culturome and bulk soils, chernozem (CZ) and sod-podzolic (SP) soil samples were significantly separated according to beta diversity metrics (PERMANOVA for bulk soils by Bray-Curtis: $R^2 = 0.75$, p-value = 0.003; culturome $R^2 = 0.39$, p-value = 0.004; the significant difference (p-value < 0.05) by weighted/unweighted UniFrac, mean pairwise distance, **Fig 3**).

Bulk soil samples were generally more diverse in terms of beta-diversity as compared to the corresponding culturomes. The distance between the centroids (the centres of the distributions) belonging to bulk soil samples was higher on the species and genus levels than for culturomes (0.241 for bulk soil samples and 0.155 for culturomes). This tendency reduces and finally turns to the opposite direction when the phylotypes are joined to the higher taxonomic ranks (e.g. on the family level, the numbers of common phylotypes were 72 for bulk soil and 17 for culturomes; the values of the distances between centroids were 0.059 and 0.09 respectively). The dynamics of the discussed changes can be seen more clearly using a graphical representation in Fig 4. The analysis was built on the calculation of the percent of the common tree leaves (tree tips) between the compared samples. The linear trend in the reduction of this value (ANOVA p-value < 0.05) was revealed when moving from the lowest to the highest taxonomic levels and until the number of the tree tips has reached 40–60. The linear regression differences expressed in the values of the line inclination (47.23 for bulk soil samples and 98.27 for culturomes) are statistically significant (p-value < 0.0001).

## Discussion

In this work, a mixed culturome and 16S amplicon approach was used to identify the soil microbiome, which allows quick screening of the microbial community from a solid nutrient medium. This combination is quite rare in the literature [41], because usually the studies are

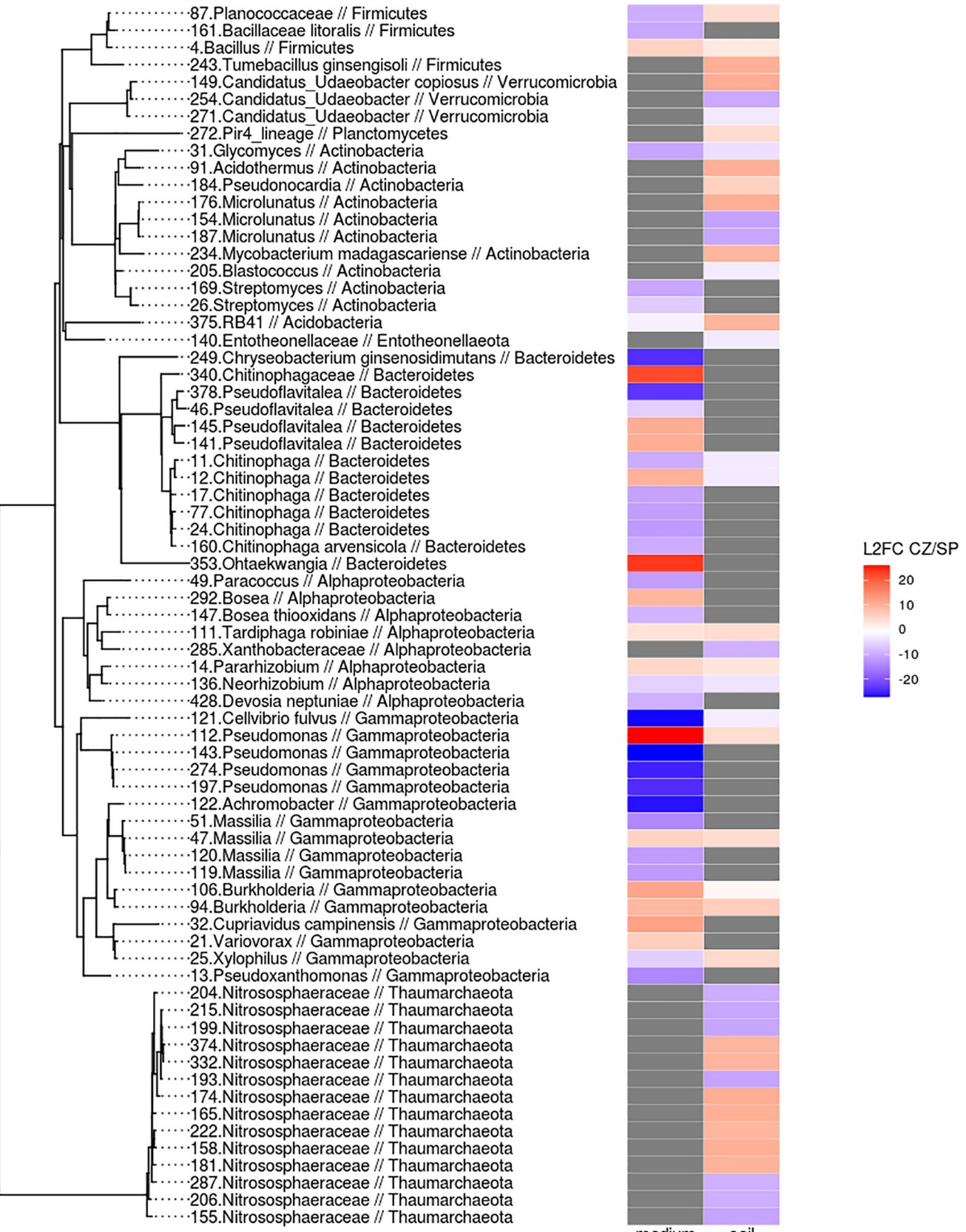

**Fig 2. The comparison of phylotype abundances for sod-podzolic soil samples (SP) vs chernozem samples (CZ) in bulk soil (soil) and culturomes (medium) samples.** Only the significant differences (padj < 0.05) within the dominant phylotypes are shown (baseMean > 60). Heat map legend meaning: increase of the phylotype abundance in CZ—red, SP—blue). The grey colour is for the phylotypes, which are absent either in bulk soil or culturome. Phylotypes not represented either in the bulk soil or in culturome are indicated in grey. CZ—chernozem soil and SP—sod-podzolic soil.

aimed at the selection and identification of individual representatives of the microbiome selected on a nutrient medium.

High inconsistency in the NGS profiles for bulk soil samples and culturomes was shown, particularly, significant number of microorganisms from the culturome were absent in the bulk soil. Moreover, no linear relationship was found in phylotypes' abundances between bulk soil samples and culturomes. This phenomenon was discussed previously by Shade and co-authors for the rhizosphere bacterial communities [8]. Probably, the most feasible source for the culturable part of the metagenome in this case is the microbial "seed bank" [42]. Other authors comparing culturome and NGS-based methods for capturing biodiversity also described a similar phenomenon. In a study of bottom sediment bacteria [43], using the enrichment method, the number of identified phylotypes increased by 16%, while the rarefaction curves almost reached the asymptote, as in our study. In a cultural study aimed at studying the soil microbiome of The Atacama Desert [41], some of the isolates were also not identified with 16S metabarcoding. In a study of the culture of the gut mouse microbiome [44], it was shown that only an insignificant part of the culture and the 16S microbiome coincide, while the culture-specific phylotypes are largely associated precisely with the functional activity of the microbiome.

However, the observed tendencies in phylotypes' distribution might be caused by the insufficient sequencing depth for bulk soil samples. So, an increase in the sample number will balance the community composition.

The difference in the taxonomic structure is characteristic when comparing the microbiome in different soil types. In our work, it was shown that the differentiation between contrast soil types for a culturome begins to appear at a higher taxonomic level than for bulk soils. This can be attributed to the fact that the selection acting while seeding microorganisms on a solid medium is manifested at a high taxonomic level. At the same time, for many microorganisms in bulk soil, soil specificity was characteristic precisely at a low taxonomic level. It should

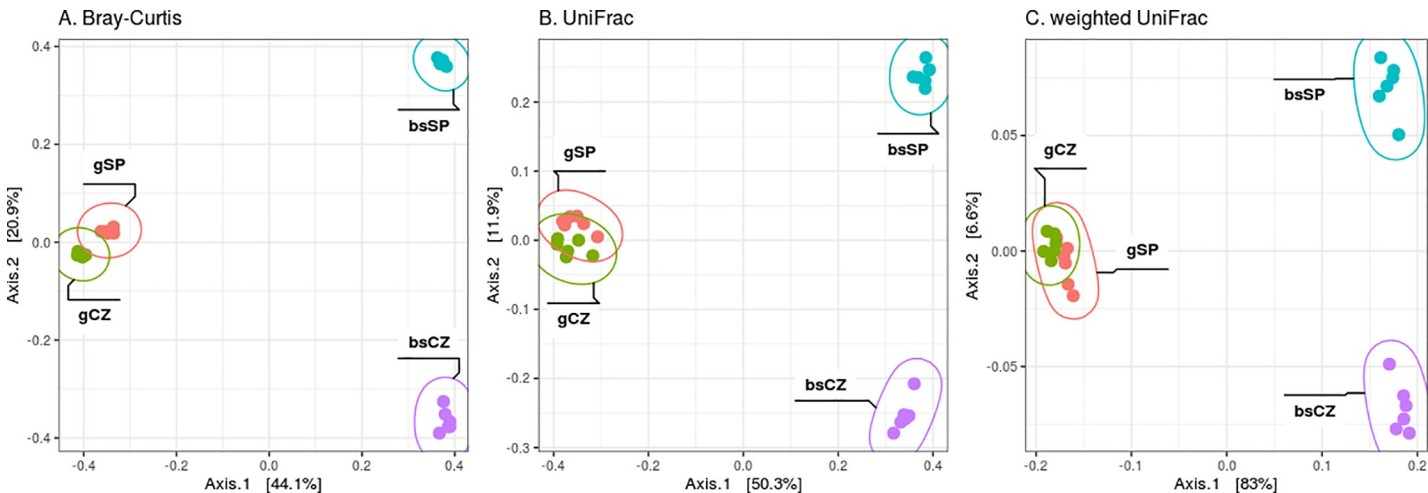

**Fig 3. PCoA ordination plots for various beta diversity metrics.** A. Bray-Curtis B. unweighted UniFrac C. weighted UniFrac. gCZ—Hutchinson medium culturome from the chernozem soil, gSP—Hutchinson medium culturome from the sod-podzolic soil, bsCZ—bulk chernozem soil, bsSP—bulk sod-podzolic soil.

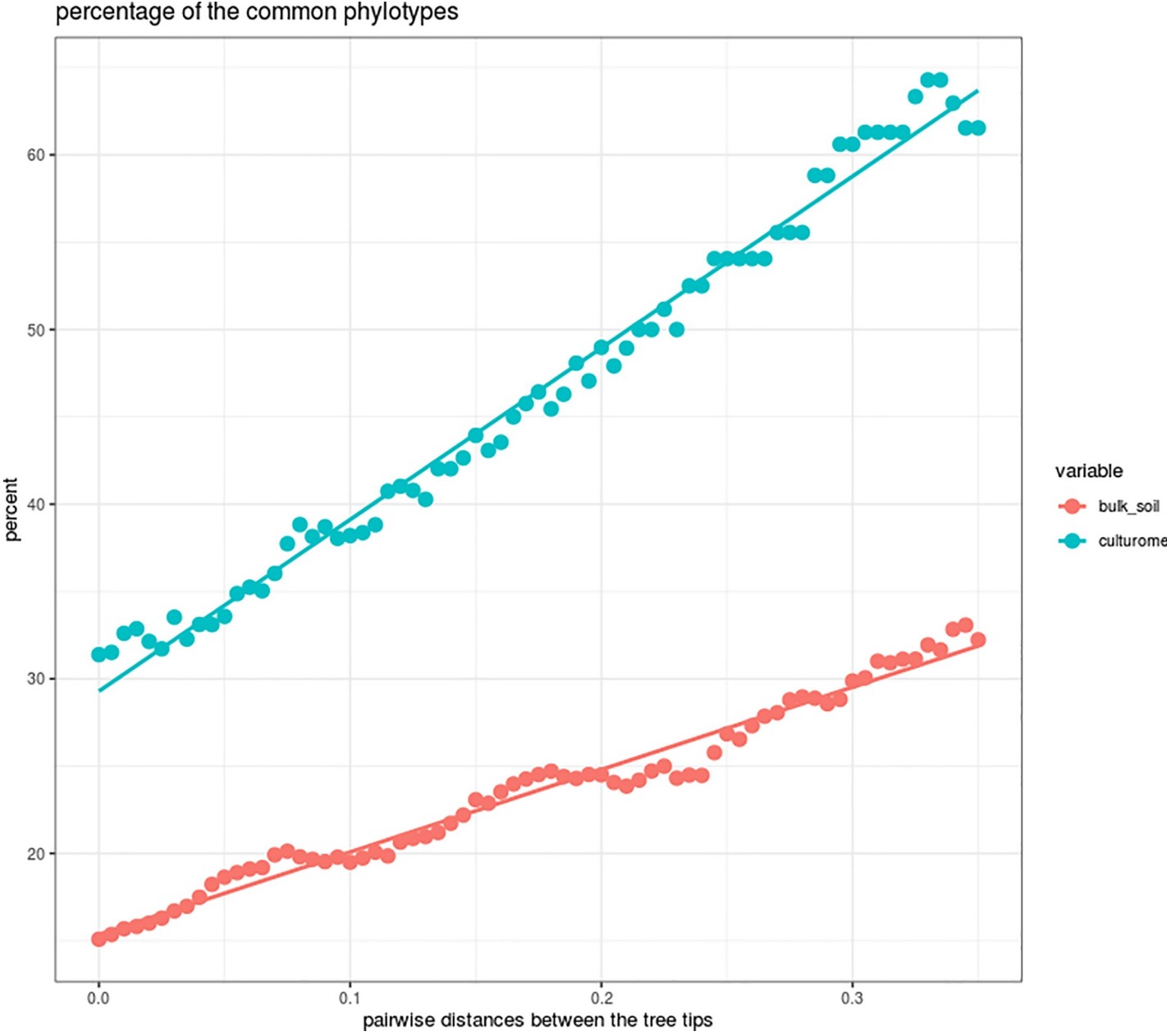

**Fig 4. Percentage of the common phylotypes inhabiting sod-podzolic soil samples (SP) and chernozem samples (CZ) measured by the calculation of the tips (leaves) of the corresponding phylogenetic trees.** A linear trend is shown by the regression line for bulk soil samples as well as for the culturomes.

be noted that there is a significant difference between the microbiomes of SP and CZ, which begins to manifest itself more strongly when growing on a solid nutrient medium that selects cellulolytic microorganisms in the soil. The results obtained may also indicate that part of the community with a low representation, but metabolically specific plays a significant role in the difference between the soil types.

Culturomes were dominated by bacteria that seemed to have copiotrophic lifestyles, e.g., some groups of beta-, gamma-, and alpha-proteobacteria, belonging to the genera *Massilia* (also known as root-colonising microbes), *Pseudomonas* and bacteria from the family

Xanthomonadaceae and Rhizobiaceae. Here we should also add actinobacteria from the genus *Glycomyces*, which were associated with plant roots [45]. These bacteria can play the role of an associated microbiota for the slow-growing and potentially cellulose-degrading bacteria, e.g. from the genus *Caulobacter* [46].

A noticeable group of bacteria within culturomes could be attributed to the cellulolytic community. Among the candidates for utilisation of complex soil polymers were bacteria belonging to the genera *Dyadobacter* [47], *Chitinophaga*, *Niastella* [48], *Flavobacterium* [49], *Cellvibrio* [50], *Steroidobacter* [51], *Stenotrophomonas* [52], *Myxococcus* [53], *Variovorax* [54], *Paenibacillus* (particularly *P. polymyxa*) [55], *Cohnella* (particularly *C. panacarvi*, which was observed in the analysed culturomes) [56], *Streptomyces* [57], *Achromobacter* [58] and *Sphingomonas*.

The microbiomes of bulk soil samples showed greater diversity than those of culturomes. Most of the species inhabiting soil microbiomes are unculturable, so little is known about their morphological features and metabolic capacities. Only a few bacteria can be partially charac-terised in this respect. Particularly, bacteria from the genus *Rubrobacter* dominated chernozem microbiomes. This bacterium, together with the large group of *Gaiellales* representatives are reported to be thermophiles, and some of them are involved in the degradation of xylan—a member of the hemicellulose group [59]. The potential for hydrolytic activity, particularly beta-glucosidase activity, was also shown for bacteria belonging to the genera *Microlunatus* [60] and *Kribbella*. In particular, *K. jejuensis* was mentioned for its utilisation of xylan and cel-lobiose [61].

All other bacteria, as well as archaea, belonged to phylogenetic groups that were discovered in the last two decades, among them were Verrucomicrobia, Thaumarchaeota, and Acidobac-teria. There is still a lack of information on their biology because many of them avoid cultiva-tion. The newest publications showed that it might be the consequence of auxotrophy [62] and potentially obligate symbiotic strategies of their lifestyle. Many of them are known to be ubiq-uitous and widespread soil bacteria, including those from the list, namely the genera RB41, *Udaeobacter*, and *Nitrososphaera*. The last one is an example of the unique ecological niche occupied by soil archaea—the indispensable link in the soil nitrogen cycle [63].

## Conclusion

The study of microbiomes together with the cultivated cellulolytic communities of two con-trasting soil types was performed by using 16S rRNA phylogenetic typing. Differences in com-munity structure between the studied soil types for bulk soils and culturomes showed that only a small part of the cellulolytic community of the culturome is identified in bulk soil. The soil- and culturome- specific soil microbiome communities were specified. The opportunity to dis-tinguish these groups proved to be very useful in studying the soil microbiome, which tends to be one of the most complex scientific subjects to date. Its complexity obliges the use of many cross-sections of biodiversity, e.g. differential DNA extraction, SIP or transcriptome analyses, which might be used to extend the current study in the future.

## Supporting information

**S1 Fig. Sampling sites visualized on a map.** Made with Natural Earth. Free vector and raster map data @ naturalearthdata.com.
(TIF)

**S1 Table. Core phylotypes for bulk soil and culturome samples.**
(CSV)

## Acknowledgments

The research was performed using equipment from the Core Centrum 'Genomic Technologies, Proteomics and Cell Biology' in All-Russia Research Institute for Agricultural Microbiology (ARRIAM). Soil sampling, laboratory experiments, high-throughput sequencing and bioinformatic analyses were supported by the Russian Scientific Foundation project RSF 18-16-00073: "Analysis of the structural and functional diversity of cellulolytic microorganisms communities using metagenomic and metatranscriptomic methods".

## Author Contributions

**Conceptualization:** Elizaveta V. Evdokimova, Grigory V. Gladkov, Evgeny E. Andronov.

**Data curation:** Elizaveta V. Evdokimova, Evgeny E. Andronov.

**Formal analysis:** Elizaveta V. Evdokimova, Grigory V. Gladkov, Natalya I. Kuzina, Ekaterina A. Ivanova, Anastasiia K. Kimeklis, Aleksei O. Zverev, Arina A. Kichko, Tatyana S. Aksenova.

**Funding acquisition:** Evgeny E. Andronov.

**Investigation:** Elizaveta V. Evdokimova, Anastasiia K. Kimeklis.

**Methodology:** Grigory V. Gladkov.

**Project administration:** Evgeny E. Andronov.

**Resources:** Alexander G. Pinaev, Evgeny E. Andronov.

**Supervision:** Evgeny E. Andronov.

**Validation:** Grigory V. Gladkov.

**Visualization:** Grigory V. Gladkov.

**Writing – original draft:** Elizaveta V. Evdokimova.

**Writing – review & editing:** Elizaveta V. Evdokimova, Grigory V. Gladkov, Evgeny E. Andronov.

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
