## [Decision Letter · Decision Letter 0]

3 Mar 2020

PONE-D-19-26379

The difference between cellulolytic ‘culturomes’ and microbiomes inhabiting two contrasting soil types

PLOS ONE

Dear Dr. Pershina,

Thank you for submitting your manuscript to PLOS ONE. After careful consideration, we feel that it has merit but does not fully meet PLOS ONE’s publication criteria as it currently stands. Therefore, we invite you to submit a revised version of the manuscript that addresses the points raised during the review process.

We would appreciate receiving your revised manuscript by Apr 17 2020 11:59PM. To enhance the reproducibility of your results, we recommend that if applicable you deposit your laboratory protocols in protocols.io, where a protocol can be assigned its own identifier (DOI) such that it can be cited independently in the future. For instructions see: http://journals.plos.org/plosone/s/submission-guidelines#loc-laboratory-protocols

We look forward to receiving your revised manuscript.

Kind regards,

Chih-Horng Kuo, Ph.D.

Academic Editor

PLOS ONE

Journal Requirements:

Reviewers' comments:

Reviewer's Responses to Questions

**Comments to the Author**

1. Is the manuscript technically sound, and do the data support the conclusions?

Reviewer #1: Partly

Reviewer #2: Yes

2. Has the statistical analysis been performed appropriately and rigorously? 

Reviewer #1: No

Reviewer #2: No

3. Have the authors made all data underlying the findings in their manuscript fully available?

Reviewer #1: Yes

Reviewer #2: Yes

4. Is the manuscript presented in an intelligible fashion and written in standard English?

Reviewer #1: No

Reviewer #2: Yes

5. Review Comments to the Author

Reviewer #1: In general, both culture independent, e.g. -Omics approaches and culture dependent methods have their own advantages and drawbacks. The intention of the current study is to integrate the advantages from both methods to reveal the difference of microbial community from two distinct types of soil and the functional groups associated with the digestion of cellulose. However, based on the information reported in the manuscript, many important components in a high quality scientific article such as the experimental design of cultivation, the purpose of using cellulose filters as source of cellulose, sampling plan, the association between the degradation of cellulose and types of soil, and methods to visualize beta diversity are not well stated. Instead, some paragraphs in the section of Introduction describing the results of metagenomic studies seem redundant.

The other concern is that the truly important findings are obscure. As stated in the manuscript, the main goal of the current study is simultaneous analysis of cellulolytic culturomes and the corresponding metagenomes in two contrasting biomes from the temperate and sub-tropical grasslands of Russia. However, the description of the results are too restricted to the description of the dominant taxa and the variation of diversity indices. The connection with the types of soils and causes of the differences in community composition is less stated and discussed in the manuscript. If the authors can link or explain the differences of microbial community composition with types of soils or environmental factors, the results will be more interesting. I also wonder the main findings or differences between the current study and published data.

Finally, the procedures and duration of cultivation were not declared in the manuscript. The authors only compared the variation between culturome and bulk soil at one point in time. If the processes associated with cellulose degradation in the cultivation and community variation in time-series could be revealed, the results will become more abundant, raising the scientific importance of this manuscript.

Here are few comments which you might wish to consider.

1. In Line 34: is there a name of genus missed after Candidatus?

2. It will be more understandable, if the style to present number is uniform in the manuscript. For example, in the text, the decimal point is indicated by a period, but in the tables, comma is used to indicate decimal place.

3. In the section of Materials and methods, the description of sampling sites and sampling strategy is unclear. I cannot understand how many samples collected in total and whether both types of soils were sampled from both Pskov and Voronezh. Based on the results present in Fig. 2, it seems that 6 samples were taken from SP and CZ soils, respectively. However, it is unclear whether for each soil type, 3 replicates were taken from Pskov and the other three were from Voronezh, or the 6 replicates were taken from the same region. I also wonder why you said soil samples were taken from 10 different equidistant points in line 115 and in the next sentence you said that “In total, 6 replicates for each type of soil were formed”. Does it mean that each replicate in each type of soil includes 10 samples from the depth of 10 cm? It will be more understandable, if you can provide a sample list which includes the coordination of sampling sites, soil type, and sampling depth, or provide a map indicating sampling sites.

4. In the part of “Bacterial Growth on nutritional medium”, I would like to know about more detail. It is not clear liquid or solid medium used for cultivation. Additionally, it is not clear the amount of soil used to cultivate and the duration of cultivation either. Please provide more information in this part.

5. In Line 175: Does the Dice methods mean “Sørensen–Dice coefficient”?

6. Please use PD whole tree and observed otus instead of the methods “PD_whole_tree” and “observed_otus” of QIIME.

7. Please uniform the usage of indices or indexes.

8. In the section of “Materials and methods”, you mentioned the indices of evenness including Faith’s index and Shannon evenness, but in “Results”, you did not show values of Faith’s index. Does PD_whole_tree means Faith’s index? If so, please uniform the name of index.

9. The authors did not state the methods for the ordination of Bray-curtis and Dice diversity estimates in Fig. 1.

10. For Fig. 2, it is suggested to use percentage instead of number of seqs.

Reviewer #2: The difference between cellulolytic ‘culturomes’ and microbiomes inhabiting two contrasting soil types

In all, attempt is good, isolating and culturing is still an important part of research. Scientific community can be happy with new isolates that can degrade cellulose. I only don’t understand the extensive comparisons of alpha and beta diversity of the culturomics with the total microbiome. It is all so logic these differences, comparing apples with peares. One or two sentences say as much. I would re-focus the results and discussion section, what was the real purpose of the study?

Line 34: genus Candidatus? Wrong..

The authors use a very selective culture media. Why they draw such comparisons between culturome phylotypes and core community? It is logic the communities differ..

Introduction

Line 48-49: a reason that media selection is straightforward, does it makes this a popular research subject? A bit weird reasoning. Sentences 50, 51 do not really connect.

Line 58: entire biodiversity: is a bit exaggerated… ‘entire’, reword. Line 59: they used 13C-SIP?

Line 82 and further: I appreciate the intend for ‘culturomics’, there is no need to defend this approach as being ‘old school’. It serves its purpose and NGS for other purpose so they can perfectly complement each other, depending on the research question. Line 87-88: combining culturing techniques with NGS, I agree it declines, but still I would not call it very few examples.

Table 1: interpret the particle size distribution please, so sandy soil? Which soil classification type? pH in water? N, P, K total or extractable elements? Many information lacking…

Line 145: picoM of primers, is little? Usually 200 nM of primers, and 200 µM of dNTPs instead of 2 nM??

Line 166: which database was used for classification and which version?

Line 164: why de novo, is not recommended by Qiime, open ref-based yes.

Line 174: beta-diversity in PAST3, bray Curtis? What about normalisation? What about other Unifrac based methods?

Line 186: MEGA X needs a reference

Line 197: wrong: the PD whole tree of culturomes cannot exceed those for bulk soils.

Table 2: in header, DP and in text, and legend SP? I cannot follow this.

Line 198: substantial decrease, significant?

Line 200: had lower, statistics done? No..

Line 216-217: so culturomics community is similar to whole bulk soil microbiome, based on presence/absence of OTUs, species, which taxonomic level similar? Hard to believe similar in species…Next line 217: they are again separate compact clusters, which statistics applied?

Line 228: OTUs which amounts…, amounts poor English, proportions, relative abundance?

Line 231: qualitative is usually which OTUs, quantitative: how much, RA of the OTUs

Line 235: explain this sentence again, not clear. In which both cases?

Line 280: I don’t know how useful it is to calculate simper analyses on cultured bacteria versus total soil sample sequencing… actually this remark holds for the whole results description. Apples comparing with pears. Keep culturomics to one paragraph, and total community seq to other. Make comparisons between the total and cultured collection, ok, but not in terms of diversity please.

Line 315-316: very logic cultured collection differs in diversity from total microbiome community composition, and in taxonomic composition and in core microbiome…

Line 318-319: very logic conclusion, no study needed for this

Line 320: might be the consequence, really might, it is obvious the consequence of…

Line 321: tiny portion… subset of…

Line 324-326: or did you sequence artifacts?

Line 328-331: again all very logic. I liked the idea of culturomics, but the results description and discussion is so straightforward and disappointing. A different angle of discussion could have been followed here, what was the real purpose of the study, isolating more cellulolytic bacteria to study their function? There is no use to extensively compare beta and alpha diversity of culturomics with total microbiome sequencing, mention it in one sentence.

Rewrite some paragraphs in the discussion. Find the message you want to give to the readers.

6. PLOS authors have the option to publish the peer review history of their article (what does this mean?). If published, this will include your full peer review and any attached files.

Reviewer #1: Yes: Tzu-Hsuan Tu

Reviewer #2: Yes: Sofie Thijs

---

## [Author Response · Author response to Decision Letter 0]

22 Jul 2020

Reviewer #1: 

At the request of reviwer #2, the structure of the work was significantly revised. Data analysis with MiSeq was re-done, the results were rewritten, the discussion was partially rewritten.

1. In Line 34: is there a name of genus missed after Candidatus?

This part of the text was rewritten.

2. It will be more understandable, if the style to present number is uniform in the manuscript. For example, in the text, the decimal point is indicated by a period, but in the tables, comma is used to indicate decimal place.

Fixed 

3. In the section of Materials and methods, the description of sampling sites and sampling strategy is unclear. I cannot understand how many samples collected in total and whether both types of soils were sampled from both Pskov and Voronezh. Based on the results present in Fig. 2, it seems that 6 samples were taken from SP and CZ soils, respectively. However, it is unclear whether for each soil type, 3 replicates were taken from Pskov and the other three were from Voronezh, or the 6 replicates were taken from the same region. I also wonder why you said soil samples were taken from 10 different equidistant points in line 115 and in the next sentence you said that “In total, 6 replicates for each type of soil were formed”. Does it mean that each replicate in each type of soil includes 10 samples from the depth of 10 cm? It will be more understandable, if you can provide a sample list which includes the coordination of sampling sites, soil type, and sampling depth, or provide a map indicating sampling sites.

The information in the Materials and methods section was changed.

4. In the part of “Bacterial Growth on nutritional medium”, I would like to know about more detail. It is not clear liquid or solid medium used for cultivation. Additionally, it is not clear the amount of soil used to cultivate and the duration of cultivation either. Please provide more information in this part.

Explanation was added

5. In Line 175: Does the Dice methods mean “Sørensen–Dice coefficient”?

Beta and alpha diversity metrics were recalculated using UniFrac, unweighted UniFrac and Bray-Curtis metrics.

6. Please use PD whole tree and observed otus instead of the methods “PD_whole_tree” and “observed_otus” of QIIME.

 Faith Index did not use in the revised version of the article?

7. Please uniform the usage of indices or indexes.

Thank. Fixed

8. In the section of “Materials and methods”, you mentioned the indices of evenness including Faith’s index and Shannon evenness, but in “Results”, you did not show values of Faith’s index. Does PD_whole_tree means Faith’s index? If so, please uniform the name of index.

this part of the work was changed, according to the request of the second reviewer, 

9. The authors did not state the methods for the ordination of Bray-curtis and Dice diversity estimates in Fig. 1.

Fixed

10. For Fig. 2, it is suggested to use percentage instead of number of seqs.

This figure have replased.

Reviewer #2: 

Introduction

Line 48-49: a reason that media selection is straightforward, does it makes this a popular research subject?

Я не знаю что здесь ответить..

This medium is often used in the institute of agricultural microbiology, where this work was done. We are looking forward to compare the obtained metagenomic data with the previous research. This medium also known to be standard for the culturing of cellulolytic microorganisms, e.g. https://sfamjournals.onlinelibrary.wiley.com/doi/pdf/10.1111/j.1365-2672.1997.tb03299.x

 A bit weird reasoning. Sentences 50, 51 do not really connect.

Line 58: entire biodiversity: is a bit exaggerated… ‘entire’, reword. Line 59: they used 13C-SIP?

This part of the introduction has been revised.

Line 82 and further: I appreciate the intend for ‘culturomics’, there is no need to defend this approach as being ‘old school’. It serves its purpose and NGS for other purpose so they can perfectly complement each other, depending on the research question. Line 87-88: combining culturing techniques with NGS, I agree it declines, but still I would not call it very few examples.

Fixed. Added to the discussion are examples of works comparing the cultural approach and the results of NGS

Table 1: interpret the particle size distribution please, so sandy soil? Which soil classification type? pH in water? N, P, K total or extractable elements? Many information lacking…

P2O5 and K2O by Machigin method, N – total. pH – in water.

Line 145: picoM of primers, is little? Usually 200 nM of primers, and 200 µM of dNTPs instead of 2 nM??

5 pM is a сommonly used concentration according to Illumina MiSeq technology standarts a

Line 166: which database was used for classification and which version? 

Data analysis with Illumina MiSeq was redone again using a different pipeline, more correct for this work. The taxonomic classification of phylotypes was determined using the RDP classifier based on Silva 132

Line 164: why de novo, is not recommended by Qiime, open ref-based yes.

This part of analysis has been redone

The whole analysis was done again by use of database-independent methods for out-picking

Line 174: beta-diversity in PAST3, bray Curtis? What about normalisation? What about other Unifrac based methods?

Redone. The main emphasis is on phylogenetic metrics.

Line 186: MEGA X needs a reference

This part has been changed

Line 197: wrong: the PD whole tree of culturomes cannot exceed those for bulk soils.

Thank

Table 2: in header, DP and in text, and legend SP? I cannot follow this.

This figure has been replased

Line 198: substantial decrease, significant?

We agree. Less emphasis is placed on alpha diversity indices.

Line 200: had lower, statistics done? No..

In the corrected work, where possible, statistics are added

Because according to the data obtained below, we cannot say with certainty that the depth of sequencing for bulk soils is sufficient; comparisons of alpha diversity indices are removed from the results. In addition, the results have been significantly reworked. Where possible, statistical processing has been added.

Line 216-217: so culturomics community is similar to whole bulk soil microbiome, based on presence/absence of OTUs, species, which taxonomic level similar? Hard to believe similar in species…Next 

This part has been rewrited

line 217: they are again separate compact clusters, which statistics applied?

ANOVA-like test for beta-diversity results has been added

Line 228: OTUs which amounts…, amounts poor English, proportions, relative abundance?

In the previous version - relative abundance

Line 231: qualitative is usually which OTUs, quantitative: how much, RA of the OTUs

Thank

Line 235: explain this sentence again, not clear. In which both cases?

This part of analysis has been completely redone

Line 280: I don’t know how useful it is to calculate simper analyses on cultured bacteria versus total soil sample sequencing… actually this remark holds for the whole results description. Apples comparing with pears. Keep culturomics to one paragraph, and total community seq to other. Make comparisons between the total and cultured collection, ok, but not in terms of diversity please.

Direct comparisons of bulk soil and culturome were removed from the article. In addition, the discussion has been revised, comparisons have been removed based on the direct use of diversity indices.

Line 315-316: very logic cultured collection differs in diversity from total microbiome community composition, and in taxonomic composition and in core microbiome…

In the present version of the work, this part is significantly reduced

Line 318-319: very logic conclusion, no study needed for this

Thank

Line 320: might be the consequence, really might, it is obvious the consequence of…

Thank. This part of discussion have been completely rewritten

Line 321: tiny portion… subset of…

Line 324-326: or did you sequence artifacts?

We cannot deny that methodological artifacts could have a significant impact on the results of the work. In a discussion in several places, an attempt was made to point this out.

Line 328-331: again all very logic. I liked the idea of culturomics, but the results description and discussion is so straightforward and disappointing. A different angle of discussion could have been followed here, what was the real purpose of the study, isolating more cellulolytic bacteria to study their function? There is no use to extensively compare beta and alpha diversity of culturomics with total microbiome sequencing, mention it in one sentence.

Rewrite some paragraphs in the discussion. Find the message you want to give to the readers.

A significant part of the analysis has been redone. Your comments have been taken into account, if possible.

---

## [Decision Letter · Decision Letter 1]

13 Aug 2020

PONE-D-19-26379R1

The difference between cellulolytic ‘culturomes’ and microbiomes inhabiting two contrasting soil types

PLOS ONE

Dear Dr. Evdokimova,

Thank you for submitting your manuscript to PLOS ONE. After careful consideration, we feel that it has merit but does not fully meet PLOS ONE’s publication criteria as it currently stands. Therefore, we invite you to submit a revised version of the manuscript that addresses the points raised during the review process.

We look forward to receiving your revised manuscript.

Kind regards,

Chih-Horng Kuo, Ph.D.

Academic Editor

PLOS ONE

Additional Editor Comments (if provided):

Dear authors,

Some minor modifications were suggested by the reviewer, please revise accordingly. Best, CH

Reviewers' comments:

Reviewer's Responses to Questions

**Comments to the Author**

1. If the authors have adequately addressed your comments raised in a previous round of review and you feel that this manuscript is now acceptable for publication, you may indicate that here to bypass the “Comments to the Author” section, enter your conflict of interest statement in the “Confidential to Editor” section, and submit your "Accept" recommendation.

Reviewer #1: All comments have been addressed

2. Is the manuscript technically sound, and do the data support the conclusions?

Reviewer #1: Yes

3. Has the statistical analysis been performed appropriately and rigorously? 

Reviewer #1: Yes

4. Have the authors made all data underlying the findings in their manuscript fully available?

Reviewer #1: Yes

5. Is the manuscript presented in an intelligible fashion and written in standard English?

Reviewer #1: No

6. Review Comments to the Author

Reviewer #1: The manuscript has been major revised based on the comments from the two reviewers. However, there are still some ambiguous places have to be clarified and improved.

Introduction

In general, the introduction provided a complete review of cellulolytic soil communities from distinct climatic zones. The taxonomic information of the dominant cellulolytic decomposers was also described in detail. However, the overall description of these paragraphs seems not compact and little bit redundant. The main purpose of the current study is to connect metagenomic approach and cellulolytic culturomes to reveal whole microbial community composition and diversity of cellulolytic decomposers in the climatic unstable temperate and subtropical grassland. Therefore, it would be better to focus on this topic and not too disperse.

L48, L59, and L82: The way to describe the first author and the remaining coauthors should be unified.

L57: phylum name is not necessary in italic form.

L70: there is a space between comma and sequencing.

Materials and Methods

It is suggested to use map to illustrate the sampling locations. Currently, it is not clear where the samples were taken.

Results

Alpha diversity of soil microbiomes and culturomes

Currently, only number of observed OTUs was used to demonstrate alpha diversity in both soil microbiomes and culturome. However, the number of observed OTUs only reflects species diversity and can not reveal the information of evenness. Therefore, it would be better to demonstrate the pattern of alpha diversity not only in number of observed OTUs, but also in other indices such as Shannon or Simpson. According to the manuscript, the term OTU is not present, but, in Figure 1A, the title of Y axis is “Number of observed OTUs”. It would be better to use phylotypes to replace OTUs.

Identification of the core and accessory components of soil microbiomes and culturomes

It seems that the authors compare two different types of soil and culturomes together to find out the shared phylotypes. It is suggested that the comparison should be separated based on the types of soils. SP and its culturome, and CZ and its culturome should be compared separately because they are different types of soils and originally have their specific core microbiome. The low number of shared phylotypes may be resulted from different types of

Please provide the information of total number of phylotypes in each sample.

L198: It is not necessary to put the family name in italic form.

L198-L200: If you would like to list the family name after each genus, it would be better to add it after all genera and at the same taxonomic level, and not some in the level of family and some in the level of order.

L200: Pactrobactertes  I have no idea what it is.

L203: Please modify “Verrucomicrobia phylum” to phylum Verrucomicrobia.

L204: Please modify “Candidatus Udaeobacter phylotypes” to Candidatus Udaeobacter phylotypes.

Fig. 3: It is suggested to add the stress values of NMDS analyses.

Discussion

L269-L272: Please add citation.

In the paragraph starts from L273 to L285, similar to the pattern in the section of Introduction, the described examples seem too disperse. It would be better to focus on similar habitats.

L289-L291: I can not fully understand the meaning of “the high degree of composition of culturome data and methodological artefacts” because the microbial composition of culturome seems simpler than bulk community. Therefore, I have no idea about the meaning of high degree of composition.

L289: Could you explain more about the meaning of “high resolution of the classical cultural approach”? Generally, classical cultural only can reflect a small proportion of microorganisms from field sample. Therefore, I have no idea why the classical cultural approach possesses high resolution.

L304: Modify Xanthomonadaceae and Rhizobiaceae family to the family Xanthomonadaceae and Rhizobiaceae

L313: Modify Paenibacillus polymyxa to P. polymyxa

L313: Modify Cohnella panacarvi to C. panacarvi

7. PLOS authors have the option to publish the peer review history of their article (what does this mean?). If published, this will include your full peer review and any attached files.

Reviewer #1: No

---

## [Author Response · Author response to Decision Letter 1]

7 Oct 2020

Below is a list of fixes based on the reviewer's answer:

L48, L59, and L82: The way to describe the first author and the remaining coauthors should be unified.

Done

L57: phylum name is not necessary in italic form.

Done

L70: there is a space between comma and sequencing.

Done

Materials and Methods

It is suggested to use map to illustrate the sampling locations. Currently, it is not clear where the samples were taken.

Map added to supplementary.

Results

Alpha diversity of soil microbiomes and culturomes

Currently, only number of observed OTUs was used to demonstrate alpha diversity in both soil microbiomes and culturome. However, the number of observed OTUs only reflects species diversity and can not reveal the information of evenness. Therefore, it would be better to demonstrate the pattern of alpha diversity not only in number of observed OTUs, but also in other indices such as Shannon or Simpson. According to the manuscript, the term OTU is not present, but, in Figure 1A, the title of Y axis is “Number of observed OTUs”. It would be better to use phylotypes to replace OTUs.

Fixed. Added Shannon Index

Identification of the core and accessory components of soil microbiomes and culturomes

It seems that the authors compare two different types of soil and culturomes together to find out the shared phylotypes. It is suggested that the comparison should be separated based on the types of soils. SP and its culturome, and CZ and its culturome should be compared separately because they are different types of soils and originally have their specific core microbiome. The low number of shared phylotypes may be resulted from different types of

Please provide the information of total number of phylotypes in each sample.

Thank you for noticing this annoying error in the article! The paragraph was rewritten, the picture was corrected, the statistics were redone. Table with general list of core phylotypes added to supplement. Fortunately, the conclusions from this work coincide with the previous(incorrect) paragraph results.

L198: It is not necessary to put the family name in italic form.

Fixed

L198-L200: If you would like to list the family name after each genus, it would be better to add it after all genera and at the same taxonomic level, and not some in the level of family and some in the level of order.

Fixed.

L200: Pactrobactertes  I have no idea what it is.

Typo – fixed

L203: Please modify “Verrucomicrobia phylum” to phylum Verrucomicrobia.

Done.

L204: Please modify “Candidatus Udaeobacter phylotypes” to Candidatus Udaeobacter phylotypes.

Done.

Fig. 3: It is suggested to add the stress values of NMDS analyses.

Thank you for your comment! Out of habit, I used NMDS for this dataset. Naturally, this is completely wrong - in this work the groups are very different from each other. I changed it to a more suitable PCoA.

Discussion

L269-L272: Please add citation.

In the paragraph starts from L273 to L285, similar to the pattern in the section of Introduction, the described examples seem too disperse. It would be better to focus on similar habitats.

Added. I'm afraid I could not find articles combining both approaches for the soil types studied in the article.

L289-L291: I can not fully understand the meaning of “the high degree of composition of culturome data and methodological artefacts” because the microbial composition of culturome seems simpler than bulk community. Therefore, I have no idea about the meaning of high degree of composition.

I removed the text about a higher degree of compositionality for a culturome data. For this paper, I use the more common normalization approach, without using specialized approaches for compositional data. You are right that for this article it makes no sense to get into this topic.

L289: Could you explain more about the meaning of “high resolution of the classical cultural approach”? Generally, classical cultural only can reflect a small proportion of microorganisms from field sample. Therefore, I have no idea why the classical cultural approach possesses high resolution.

Rewrote, I hope this will make the thought more accessible to the reader.

L304: Modify Xanthomonadaceae and Rhizobiaceae family to the family Xanthomonadaceae and Rhizobiaceae

Fixed

L313: Modify Paenibacillus polymyxa to P. Polymyxa

Fixed

L313: Modify Cohnella panacarvi to C. Panacarvi

Done

Thanks for your time! Your work helped us a lot.

---

## [Decision Letter · Decision Letter 2]

27 Oct 2020

The difference between cellulolytic ‘culturomes’ and microbiomes inhabiting two contrasting soil types

PONE-D-19-26379R2

Dear Dr. Evdokimova,

We’re pleased to inform you that your manuscript has been judged scientifically suitable for publication and will be formally accepted for publication once it meets all outstanding technical requirements.

Kind regards,

Chih-Horng Kuo, Ph.D.

Academic Editor

PLOS ONE

Additional Editor Comments (optional):

Reviewers' comments:

Reviewer's Responses to Questions

**Comments to the Author**

1. If the authors have adequately addressed your comments raised in a previous round of review and you feel that this manuscript is now acceptable for publication, you may indicate that here to bypass the “Comments to the Author” section, enter your conflict of interest statement in the “Confidential to Editor” section, and submit your "Accept" recommendation.

Reviewer #1: All comments have been addressed

2. Is the manuscript technically sound, and do the data support the conclusions?

Reviewer #1: Yes

3. Has the statistical analysis been performed appropriately and rigorously? 

Reviewer #1: Yes

4. Have the authors made all data underlying the findings in their manuscript fully available?

Reviewer #1: Yes

5. Is the manuscript presented in an intelligible fashion and written in standard English?

Reviewer #1: Yes

6. Review Comments to the Author

Reviewer #1: (No Response)

7. PLOS authors have the option to publish the peer review history of their article (what does this mean?). If published, this will include your full peer review and any attached files.

Reviewer #1: No

---

## [Editor Report · Acceptance letter]

11 Nov 2020

PONE-D-19-26379R2 

The difference between cellulolytic ‘culturomes’ and microbiomes inhabiting two contrasting soil types 

Dear Dr. Evdokimova:

I'm pleased to inform you that your manuscript has been deemed suitable for publication in PLOS ONE. Congratulations! Your manuscript is now with our production department. 

Kind regards, 

on behalf of

Dr. Chih-Horng Kuo 

Academic Editor

PLOS ONE